# Trends in Clinico-Epidemiological Profile and Outcomes of Patients with HIV-Associated Cryptococcal Meningitis in Shanghai, China, 2013–2023

**DOI:** 10.3390/v16081333

**Published:** 2024-08-21

**Authors:** Zihui Zhao, Wei Song, Li Liu, Tangkai Qi, Zhenyan Wang, Yang Tang, Jianjun Sun, Shuibao Xu, Junyang Yang, Jiangrong Wang, Jun Chen, Renfang Zhang, Yinzhong Shen

**Affiliations:** 1Shanghai Institute of Infectious Disease and Biosecurity, Fudan University, Shanghai 200032, China; 22211020041@m.fudan.edu.cn; 2Shanghai Public Health Clinical Center, Fudan University, Shanghai 201508, China

**Keywords:** HIV/AIDS, cryptococcal meningitis, epidemiology, clinical characteristics, prognosis

## Abstract

The study aimed to analyze changes in the clinical and epidemiological aspects of HIV-associated cryptococcal meningitis (CM) patients and to identify factors influencing their prognosis. Clinical data of patients with HIV-associated CM treated in Shanghai, China between 2013 and 2023 were collected. This study included 279 cases, 2.89% of AIDS patients, showing a yearly decrease in CM prevalence among AIDS patients (*p* < 0.001). Overall mortality was 10.39% with rates declining from a 2013 peak of 15.38% to 0% in 2023 despite no significant temporal pattern (*p* = 0.265). Diagnosis took an average of 18 ± 1 days post-symptoms, and admission CD4 counts averaged 29.2 ± 2.5 cells/μL, hinting at a non-significant decline. Frequent symptoms included fever (62.4%), headache (61.6%), fatigue (44.1%), and appetite loss (39.8%), with younger patients more likely to initially show signs of meningeal irritation. Logistic regression analysis underscored the prognostic importance of cerebrospinal fluid (CSF) white blood cell (WBC) count and procalcitonin levels. Over the decade spanning from 2013 to 2023, the incidence and mortality rates of CM among AIDS patients exhibited a downward trend. The average duration from the onset of CM to confirmation of diagnosis remained prolonged. CSF WBC count and procalcitonin levels were associated with unfavorable outcomes.

## 1. Introduction

Cryptococcosis is one of the most common opportunistic infections in patients with human immunodeficiency virus (HIV) infection and acquired immune deficiency syndrome (AIDS), with notably high incidence and mortality rates in sub-Saharan Africa. Currently, the most common clinical manifestation of cryptococcosis is cryptococcal meningitis (CM). Epidemiological studies reveal that around 152,000 cases of CM occur in HIV-infected and AIDS patients worldwide each year, resulting in nearly 112,000 deaths annually [1]. Over the past two decades, the population of CM patients has expanded beyond HIV-infected individuals to include those with potential immune deficiencies, which is largely due to the widespread clinical application of antiretroviral therapy (ART) and the increasing number of patients using immunosuppressants and organ transplant recipients [2]. Additionally, *Cryptococcus neoformans* tops the World Health Organization’s (WHO) Fungal Priority Pathogens List (FPPL), emphasizing the significance of monitoring clinical outcomes and antifungal drug resistance [3]. Despite several clinical studies of CM being published in the last eight years [4,5,6,7], there are few reports focused on people living with HIV (PLWH), and there are a lack of data on yearly variations. A recent study also found that even individuals with high CD4^+^ lymphocyte counts in PLWH populations remain vulnerable to *C. neoformans* infection [8]. Furthermore, Chen et al. illustrated that CM cases among China’s HIV-infected population may have been severely under-reported [9]. To address these concerns, this retrospective study delved into the clinical and epidemiological features of 279 confirmed cases of CM through positive cerebrospinal fluid (CSF) cultures in AIDS patients managed at the Shanghai Public Health Clinical Center (SPHCC), Fudan University, as well as yearly changing trends. The objective was to furnish insights that could guide the refinement of diagnostic and therapeutic approaches in clinical settings.

## 2. Materials and Methods

### 2.1. Study Subjects

This study conducted a retrospective analysis of patient data from the SPHCC. All patients met the AIDS diagnosis criteria, which were HIV infection with CD4^+^ lymphocyte counts below 200/μL or having other conditions described in the Chinese guidelines for the diagnosis and treatment of HIV infection/AIDS [10]. HIV infection was confirmed by nuclear acid test and HIV viral load test. During hospitalization, all patients suspected of having a central nervous system infection, indicated by clinical manifestations such as headache and fever, or with positive blood cryptococcal antigen tests, underwent lumbar puncture. CSF cultures were then performed to confirm the diagnosis of CM. The treatment strategy follows the Chinese guidelines mentioned above [10]. All patients received low-dose amphotericin B (starting from 0.02 to 0.10 mg·kg^−1^·d^−1^ and gradually increasing the dose to 0.50 to 0.70 mg·kg^−1^·d^−1^) administered intravenously, which was combined with oral flucytosine (100 mg·kg^−1^·d^−1^) as the induction therapy regimen for at least 4 weeks. Following this, fluconazole (400 mg/d orally) was used for consolidation therapy until the patient’s CSF smear turned negative. Maintenance therapy with fluconazole (200 mg/d orally) was continued until the CD4^+^ T lymphocyte count was greater than 100 cells/μL, after which the medication was stopped. For precise species identification of the clinical isolates, Matrix-Assisted Laser Desorption/Ionization Time-of-Flight Mass Spectrometry (MALDI-TOF MS) was employed using the BRUKER microflex^®^ platform. We extracted patients’ initial laboratory assessments at admission or during outpatient visits, and their medical records were based on the positive culture results of CSF between January 2013 and December 2023. Each patient was included only once for the same episode of CM (see Figure 1 for the exclusion process).

### 2.2. Information Gathering

Following the identification of the study cohorts, demographic details, clinical manifestations, and additional pertinent information were extracted from the electronic medical records system. Initial laboratory or outpatient examination results were retrieved based on the patients’ admission number. The study was approved by the SHPCC Ethics Committee (No: 2021-S051-03).

### 2.3. Statistical Analysis

Patient data were arranged in Microsoft Excel, and statistical analyses were conducted using SPSS version 29.0 and R 4.4.0. GraphPad Prism version 10 was employed for graphing. Count data were expressed as frequency (percentage) and compared between groups using the χ^2^ test. For continuous variables, the Mann–Whitney U test was used for comparisons between two groups, while the Kruskal–Wallis test was used for multiple group comparisons. Firth’s Penalized Logistic Regression (R package logistf v1.26.0) was performed with *p* < 0.05 considered statistically significant. The study also employs Spearman’s correlation to analyze the relationships between variables, and the results are presented in Table 1 and Table 2.

Patient age was stratified according to the classification standards in the Centers for Disease Control and Prevention (CDC) “HIV Surveillance Report” [11], which were divided into five groups as detailed in Table 1. Cryptococcal drug sensitivity results were determined based on the Epidemiological Cutoff Value (ECV) for Cryptococcus species outlined in the Clinical and Laboratory Standards Institute (CLSI) document M57SEd4 [12] with an ECV of 8 μg/mL for *C. neoformans*.

## 3. Results

### 3.1. Baseline Characteristics

Basic information on all patients is presented in Table 1. All isolates from CSF cultures were confirmed to be *C. neoformans* using MALDI-TOF MS. A total of 279 cases that met the inclusion criteria, representing 2.89% (279/9658) of all AIDS patients, were included in the study. The prevalence of CM among AIDS patients showed a yearly declining trend (Appendix A) [χ^2^ = 15.209, *p* < 0.001]. The overall mortality rate was 10.39% (29/279). The annual mortality rate fluctuated around 13% before 2021, peaking in 2013 at 15.38% (4/26), and dropped to 0% in 2023. Despite the drastic changes in data over the last two years, the linear correlation line showed a downward trend (Appendix A). However, the trend Chi-square test results showed no significant trend in mortality over the years [χ^2^ = 1.242, *p* = 0.265].

The average time from illness onset to final CM diagnosis was 18 ± 1 days. The average hospitalization time was 49 ± 2 days, showing a yearly decrease [H = 26.707, *p* = 0.003]. The average count of CD4^+^ T lymphocytes at admission was 29.2 ± 2.5 cells/μL, showing a declining correlation over the years [r_s_ = −0.055] but no significant trend [H = 15.204, *p* = 0.125]. CSF pressure has shown a slight decrease in recent years [r_s_ = −0.055], which is statistically significant [H = 19.520, *p* = 0.034]. EBV and HBV infections varied significantly with an upward trend in EBV [χ^2^ = 41.438, *p* < 0.001, r_s_ = 0.396] and a downward trend in HBV [χ^2^ = 7.741, *p* = 0.005, r_s_ = −0.170].

### 3.2. Initial Clinical Presentation of CM Patients

Appendix A displays the most common clinical presentations reported by patients during their first consultation. Fever was the most frequent presenting complaint, which was observed in 174 patients (174/279, 62.4%). Its occurrence rate varied over the years with no significant trend [χ^2^ = 2.482, *p* = 0.115]. Following fever, the most common complaints were headache (172/279, 61.6%), fatigue (123/279, 44.1%), and poor appetite (111/279, 39.8%). The proportion of patients reporting headache showed a significant decreasing trend [χ^2^ = 4.610, *p* = 0.032, r_s_ = −0.123].

Although dizziness [χ^2^ = 8.869, *p* = 0.003] and insomnia [χ^2^ = 6.794, *p* = 0.009] reached statistically significant levels and both exhibited increasing trends, a detailed examination of the annual incidence rates reveals significant fluctuations in 2019 and 2020. These fluctuations notably impacted the overall trend.

Other less frequent clinical manifestations (with a prevalence below 5%) included seizures, shortness of breath, convulsions, decreased vision, weakness in the lower limbs, chest tightness, back pain, rash, urinary incontinence, neck pain, abdominal pain, diarrhea, hearing loss, and difficulty in defecation. Figure 2 illustrates those signs of meningeal irritation, such as headache and vomiting, were more likely to appear in the initial clinical presentations of younger patients.

### 3.3. Univariate Analysis of Prognosis Factors

In Table 2, among all 279 patients, factors associated with outcomes included CSF white blood cell (WBC) count [*p* = 0.031] with lower counts [r_s_ = −0.129] observed in the deceased group compared to the survival group. The CD3 cell count [*p* = 0.048] also reached a significant level with lower counts [r_s_ = −0.122] in the deceased group. Procalcitonin (PCT) [*p* < 0.001] is higher in the deceased group. Among overlapping infections, there were no significant differences in the prognosis for infections with *Epstein–Barr virus* (EBV), *Cytomegalovirus* (CMV) and *hepatitis B virus* (HBV). There was also no significant difference in the number of days to diagnosis of CM between these two groups.

Although the CD4 cell count [*p* = 0.084] and CD8 cell count [*p* = 0.066] did not reach the significant level, they showed a trend toward significance with lower counts [CD4 r_s_ = −0.107, CD8 r_s_ = −0.113] in the deceased group.

The impact of cryptococcal drug sensitivity results on prognosis was also analyzed. The analysis indicated that drug sensitivity had no significant impact on the outcomes, as detailed in Table 3.

### 3.4. Multivariate Logistic Regression Analysis of Prognostic Factors

The model presented in Appendix A was derived through Firth’s penalized logistic regression analysis. We selected parameters that reached a weak significance level (*p* < 0.1) in the entire dataset. The WBC count in CSF [OR = 0.908, *p* = 0.015] and PCT [OR = 1.593, *p* = 0.001] reached significant levels. The receiver operating characteristic curve (ROC) of the model is shown in Appendix A, with an area under the curve (AUC) of 0.8494, which is significantly different (*p* = 0.004).

## 4. Discussion

This study analyzed the trends in clinico-epidemiological profile, as well as the factors influencing outcomes of patients with HIV-associated CM in Shanghai, China, during the past 10 years. Our findings showed a decline in both the prevalence and mortality rates of CM among AIDS patients. The length of hospitalization time since 2013 has varied significantly, showing an overall decreasing trend. However, this does not necessarily indicate improved efficacy in CM treatment, as hospitalization duration is influenced by various factors, including patients’ own decisions. Notably, the prevalence of CM in our study population was 2.89%, and the mortality rate remained high at 10.39%. Based on the 2020 global burden model for HIV-associated cryptococcal disease in adults [1], an estimate indicated that 4.3 million adults were living with HIV and had CD4 counts below 200 cells/μL. An examination of data from UNAIDS and population-based surveys disclosed a global mean prevalence of cryptococcal antigenemia at 4.4% within this high-risk population. This translated to an approximate 152,000 cases of CM, representing about 3.53% of AIDS cases in 2020, and resulted in 112,000 deaths annually, accounting for 19% of all AIDS-related fatalities. The burden has decreased marginally since 2014 [13], which may be indicative of the effectiveness of contemporary medical interventions and advancements in HIV prevention and control efforts. The widespread implementation of early diagnosis combined with ART has facilitated the swift detection and management of HIV, thereby efficaciously diminishing the likelihood of opportunistic infections like CM in HIV-positive individuals, as reported by studies [14,15]. Nonetheless, epidemiological data imply [16] a paradoxical situation: despite enhanced accessibility to ART in many African regions with high incidence, no clear decline in case numbers of CM has been observed. This phenomenon can be attributed to a substantial proportion, nearly 50%, of CM patients having previously undergone ART but still encountering a persistent decrease in CD4^+^ T cell counts. This decline is attributed to issues like patient attrition, inadequate adherence, or the emergence of HIV resistance strains [16]. Notably, a 2017 report emphasized that even with ART availability, 20–25% of clinic visits by individuals with HIV infection still featured CD4^+^ T cell counts below 100/μL [13], and cryptococcal disease persists as the second major cause of HIV-related mortality.

In 2022, the WHO categorized *C. neoformans* as a foremost fungal priority pathogen due to its persistently high mortality rate, even amidst the use of ART [3]. Renowned for instigating disseminated infections that often affect multiple organs, this organism leads to severe complications, notably elevated intracranial pressure requiring surgical intervention, blindness, and a particular propensity for HIV-infected individuals [3,17]. In response, a mounting corpus of research and clinical guidelines has underscored the supreme significance of expeditious diagnosis as a cornerstone strategy to reduce fatalities [13,15,16]. In alignment with this emphasis, our investigation delved into the timeframe from symptom onset to definitive diagnosis alongside the CD4^+^ T cell count upon hospital admission. We found the mean interval between symptom initiation and confirmed diagnosis of CM to be 18 ± 1 days. Furthermore, the mean CD4^+^ T lymphocyte count at admission was recorded at 29.2 ± 2.5 cells/μL, which was indicative of a downward trend over recent times. Notably, however, our study failed to disclose significant enhancements in diagnostic promptness. This finding, when coupled with the statistic that 93.5% (261/279) of our enrolled participants presented with CD4 counts under 200 cells/μL, underscores the persistent struggle in achieving timely diagnostic intervention.

Regarding clinical characteristics, this study observed that the initial presentation of cryptococcal meningitis (CM) cases was largely non-specific. However, certain trends emerged: notably, an increased prevalence of symptoms such as headache, dizziness, and insomnia (Appendix A). An age-stratified analysis revealed a higher occurrence of meningeal irritation symptoms, such as headache and vomiting, among younger patients. These insights suggest that clinicians must be particularly vigilant for the sudden onset of fever and headache in individuals with HIV/AIDS, as older patients may exhibit less overt symptoms. Consequently, regular screening and diligent follow-up are advisable for the elderly population, reinforcing the recommendations of several clinical studies and guidelines [18,19,20,21,22]. These sources consistently highlight fever and headache as cardinal symptoms, with up to 20% of patients experiencing changes in mental status, while seizures and strokes are comparatively rarer [22].

In the univariate analysis, this study identified several factors influencing prognosis. Notably, the deceased group exhibited significantly lower counts in various parameters, including WBC count in CSF, CD3 count, PCT, and ART initiation (Table 2). Based on these observations, we employed Firth’s penalized logistic regression analysis to further scrutinize these variables and identified two factors with statistical significance: WBC count in CSF and PCT (Appendix A). Consistent with a 2020 publication [23], our findings underscore that a reduced WBC count in CSF is correlated with an unfavorable prognosis. This prior study suggested that CM patients presenting with lower baseline CSF WBC counts, CSF protein concentrations, or CD4/CD8 ratios are at heightened risk for poor clinical outcomes [23]. While we also investigated other parameters highlighted in the study, neither the CSF protein concentration nor CD4/CD8 ratio attained statistical significance in our analysis (Table 2). The study further explored the correlation between the top three recognized opportunistic pathogens and adverse patient outcomes. Remarkably, the results indicated no significant disparities (Table 2). Elevated PCT levels are associated with adverse outcomes, as revealed by both univariate and logistic regression analysis. In PLWH, PCT has been identified as a marker of bacterial sepsis with early evidence published in the *Journal of Infection* in 1997 [24]. A study published in 2014 found that elevated serum PCT (>0.5 ng/mL) is an independent predictor of in-hospital mortality in HIV-infected Ugandan adults with lower respiratory tract infections [25]. Additionally, a study identifying predictors of mortality among HIV-associated *Talaromyces marneffei* patients also found that PCT ≥ 1.7 ng/mL was an independent risk factor for 24-week mortality [26]. However, there are contradictory conclusions regarding PCT as a marker in HIV populations. For instance, in a South African cohort of asymptomatic, antiretroviral therapy-naive HIV-infected individuals in 2016, PCT levels were not elevated despite persistent inflammation, suggesting that PCT may not be a reliable marker for inflammation in this context [27]. Comparing our study with the South African cohort, our patients’ status is similar with the majority having no ART initiation. We also analyzed PCT with cut-offs (Appendix A) and found significant differences between the two groups. However, it is noteworthy that PCT levels in the deceased group varied dramatically. Therefore, while further research is needed to determine whether PCT can reliably serve as an indicator of severe cases, patients with high baseline PCT values should be closely monitored to prevent complications from serious infections.

Regarding drug resistance, our investigation revealed non-wild-type resistance proportions for flucytosine, amphotericin B, and fluconazole in *C. neoformans* at 1.5%, 3.8%, and 5.7%, respectively. However, our findings did not indicate a statistically significant correlation between drug sensitivity and patient outcomes. In vitro drug susceptibility testing in this study consistently showed that *C. neoformans* generally responds favorably to medications with no widespread reports of drug resistance emerging over the past decade. Consequently, isolates exhibiting elevated minimum inhibitory concentrations (MICs) in vitro did not significantly impact patient outcomes. Notably, the existing literature on *C. neoformans* drug resistance is sparse and inconsistent across various studies. A study conducted in southwest China [28] documented non-wild-type frequencies among 130 *C. neoformans* isolates as 8.5%, 8.5%, and 6.2% for these agents. Conversely, an Iranian study [29], involving 16 isolates, found no instances of non-wild-type strains. Bassetti et al., in their assessment of fluconazole resistance in *C. neoformans*, did not identify any resistant isolates in South Africa, Asia, or the Western Pacific region [30]. Nevertheless, another investigation [31] from Taiwan, China revealed that from 2001 to 2012, 34% (30/89) of *C. neoformans* isolates demonstrated reduced fluconazole susceptibility, surpassing rates previously documented. Therefore, the ongoing surveillance of clinical drug resistance in *C. neoformans* is crucial given the evident regional disparities in findings.

This study had several limitations, including the absence of data on incidence rates and recurrence monitoring for patients. Furthermore, in exploring pathogenic mechanisms, the investigation falls short in categorizing and analyzing different serotypes of *C. neoformans*.

In summary, although the prevalence of CM among HIV/AIDS patients has decreased, the mortality remains at a relatively high level. Due to the atypical clinical presentation of this disease, early identification and diagnosis are still challenging. It is crucial to strengthen the follow-up and screening of HIV/AIDS patients, especially among the middle-aged population with CD4^+^ T lymphocyte counts below 100/μL, to ensure they receive ART as early as possible. The clinical manifestations of CM in elderly HIV/AIDS patients are more concealed compared to younger individuals, making intensified follow-up and screening key in preventing this high-risk group from developing the disease. Patients with lower CSF WBC counts and higher PCT levels at admission should be monitored for potential progression to severe conditions. Although no significant in vitro antifungal drug resistance events have been observed in our hospitalized patients, regional data may vary considerably. Moreover, since current first-line treatment regimens still rely on various antifungal drugs, the close monitoring of pathogenesis and the emergence of resistant strains is essential.

## Figures and Tables

**Figure 1 viruses-16-01333-f001:**
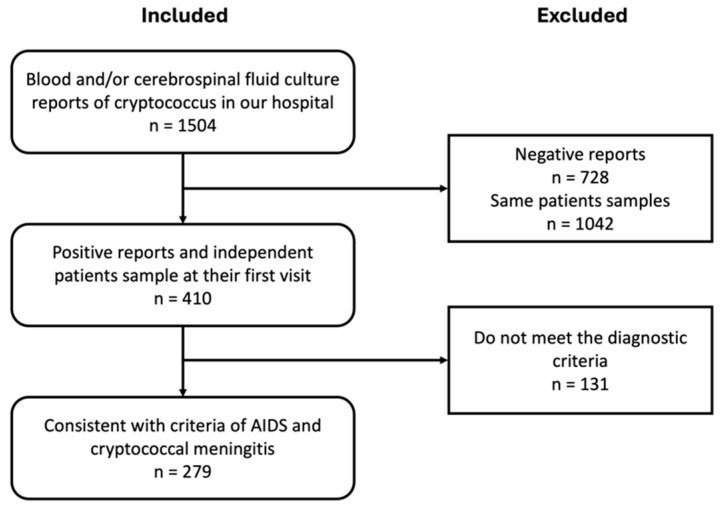
Study population flow chart. The study included 1504 positive reports, of which 279 cases met the inclusion criteria.

**Figure 2 viruses-16-01333-f002:**
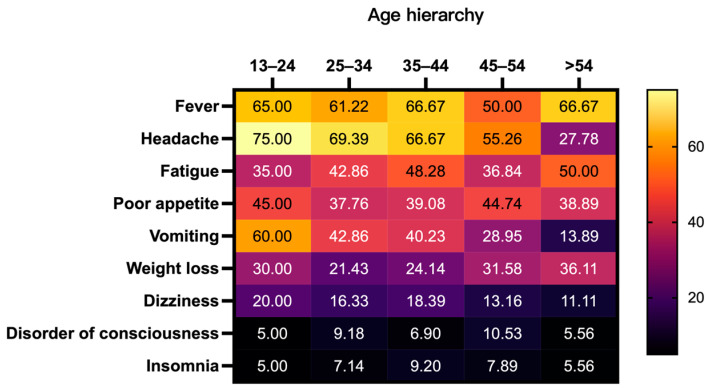
Clinical presentation heatmap stratified by age. The table displays the percentage of individuals experiencing various symptoms across different age groups. Each cell in the table indicates the percentage of individuals within the respective age group who reported the symptom, which was calculated as (number of individuals with the symptom/total number of individuals in the age group) × 100%.

**Table 1 viruses-16-01333-t001:** Baseline characters of patients with HIV-associated cryptococcal meningitis.

Parameters		Years											Subtotal	Spearman	*p* Values
		2013	2014	2015	2016	2017	2018	2019	2020	2021	2022	2023			
Gender															
	Male	23 (9.35%)	27 (10.98%)	29 (11.79%)	31 (12.6%)	25 (10.16%)	26 (10.57%)	19 (7.72%)	26 (10.57%)	23 (9.35%)	12 (4.88%)	5 (2.03%)	246	0.055	0.362 ^†^
	Female	3 (9.09%)	2 (6.06%)	2 (6.06%)	6 (18.18%)	3 (9.09%)	3 (9.09%)	4 (12.12%)	4 (12.12%)	3 (9.09%)	1 (3.03%)	2 (6.06%)	33		
Age (Years)															
	<13	0 (0%)	0 (0%)	0 (0%)	0 (0%)	0 (0%)	0 (0%)	0 (0%)	0 (0%)	0 (0%)	0 (0%)	0 (0%)	0	0.051	0.658 ^‡^
	13–24	1 (5%)	4 (20%)	2 (10%)	3 (15%)	3 (15%)	1 (5%)	1 (5%)	0 (0%)	3 (15%)	1 (5%)	1 (5%)	20		
	25–34	9 (9.18%)	10 (10.2%)	12 (12.24%)	14 (14.29%)	12 (12.24%)	−8 (8.16%)	7 (7.14%)	13 (13.27%)	9 (9.18%)	3 (3.06%)	1 (1.02%)	98		
	35–44	7 (8.05%)	9 (10.34%)	12 (13.79%)	10 (11.49%)	5 (5.75%)	16 (18.39%)	3 (3.45%)	7 (8.05%)	8 (9.2%)	6 (6.9%)	4 (4.6%)	87		
	45–54	7 (18.42%)	3 (7.89%)	3 (7.89%)	2 (5.26%)	5 (13.16%)	3 (7.89%)	9 (23.68%)	2 (5.26%)	2 (5.26%)	2 (5.26%)	0 (0%)	38		
	>54	2 (5.56%)	3 (8.33%)	2 (5.56%)	8 (22.22%)	3 (8.33%)	1 (2.78%)	3 (8.33%)	8 (22.22%)	4 (11.11%)	1 (2.78%)	1 (2.78%)	36		
ART initiation															
	Yes	4 (13.79%)	3 (10.34%)	6 (20.69%)	5 (17.24%)	3 (10.34%)	3 (10.34%)	0 (0%)	2 (6.9%)	2 (6.9%)	0 (0%)	1 (3.45%)	29	−0.120	0.051 ^†^
	No	22 (8.8%)	26 (10.4%)	25 (10%)	32 (12.8%)	25 (10%)	26 (10.4%)	23 (9.2%)	28 (11.2%)	24 (9.6%)	13 (5.2%)	6 (2.4%)	250		
Outcomes															
	Survival	22 (8.8%)	26 (10.4%)	27 (10.8%)	32 (12.8%)	26 (10.4%)	27 (10.8%)	21 (8.4%)	26 (10.4%)	24 (9.6%)	12 (4.8%)	7 (2.8%)	250	−0.066	0.266 ^†^
	Death	4 (13.79%)	3 (10.34%)	4 (13.79%)	5 (17.24%)	2 (6.9%)	2 (6.9%)	2 (6.9%)	4 (13.79%)	2 (6.9%)	1 (3.45%)	0 (0%)	29		
Identified pathogens ^§^															
	EBV	0 (0%)	0 (0%)	0 (0%)	0 (0%)	1 (2.38%)	8 (19.05%)	6 (14.29%)	11 (26.19%)	12 (28.57%)	4 (9.52%)	0 (0%)	42	0.396	<0.001 ^†^***
	CMV	1 (3.33%)	9 (30%)	5 (16.67%)	2 (6.67%)	2 (6.67%)	1 (3.33%)	0 (0%)	3 (10%)	5 (16.67%)	1 (3.33%)	1 (3.33%)	30	−0.055	0.439 ^†^
	HBV	5 (29.41%)	3 (17.65%)	2 (11.76%)	3 (17.65%)	0 (0%)	2 (11.76%)	0 (0%)	0 (0%)	2 (11.76%)	0 (0%)	0 (0%)	17	−0.170	0.005 ^†^**
	Treponema pallidum	0 (0%)	0 (0%)	2 (15.38%)	0 (0%)	4 (30.77%)	1 (7.69%)	1 (7.69%)	0 (0%)	3 (23.08%)	2 (15.38%)	0 (0%)	13	0.106	0.086 ^†^
Days from illness onset to final diagnosis (Days)		15 ± 2	14 ± 1	18 ± 2	17 ± 2	18 ± 3	15 ± 2	16 ± 3	23 ± 4	19 ± 3	22 ± 8	14 ± 3	18 ± 1	0.035	0.903 ^‡^
Hospitalization time (day)		70 ± 8	62 ± 6	48 ± 5	41 ± 4	54 ± 6	50 ± 6	36 ± 4	50 ± 5	45 ± 6	37 ± 7	35 ± 2	49 ± 2	−0.242	0.003 ^‡^**
CD4 count at admission (cell/μL)		35.7 ± 10.3	25.1 ± 3.8	31.3 ± 8.1	34.7 ± 6.3	31.9 ± 6.9	25.5 ± 13.9	25.8 ± 3.8	21.7 ± 3.4	32.5 ± 9	29.5 ± 8.4	16.9 ± 5.1	29.2 ± 2.5	−0.055	0.125 ^‡^

Data are expressed as n (%), mean ± standard error of mean or median (interquartile range). Results are based on non-empty rows and columns in each innermost subtable. ^†^. Significance value of trend Chi-square test. ^‡^. Significance value of Kruskal–Wallis non-parametric test. **. significant at the 0.01 level. ***. significant at the 0.001 level. ^§^. Identified pathogens refer to those found in patients’ blood or other sample cultures, excluding contamination and colonization. AIDS: acquired immunodeficiency syndrome. ART: antiretroviral therapy. EBV: Epstein–Barr virus. CMV: cytomegalovirus. HBV: hepatitis B virus.

**Table 2 viruses-16-01333-t002:** Factors related to outcomes in patients with HIV-associated cryptococcal meningitis.

Parameters		Outcome		*p* Value	Spearman
		Survival	Death		
Age (year)		37 (30–45)	39 (29–52)	0.694 ^†^	0.024
Gender	Male	222 (90.24%)	24 (9.76%)	0.360 ^§^	0.057
BDG (pg/mL)		25.63 (1–166.3)	58.04 (1–236.7)	0.307 ^†^	0.067
WBC in CSF (10^6 cell/μL)		4 (0.8–12)	0.5 (0.2–4)	0.031 ^†^*	−0.129
RBC in CSF (10^6 cell/μL)		0 (0–2.8)	0 (0–2)	0.763 ^†^	−0.018
Chloride in CSF (mmol/L)		119.15 (116–123)	120 (113.45–124)	0.990 ^†^	0.001
Glucose in CSF (mmol/L)		2.52 (1.85–3.03)	2.31 (1.51–3.52)	0.963 ^†^	−0.003
Protein in CSF (mg/L)		417.55 (219.7–629.5)	396 (237.6–752)	0.516 ^†^	0.039
CSF opening pressure		270 (160–355)	315 (207.5–400)	0.128 ^†^	0.092
CD3 count (cell/μL)		343 (216.67–567.66)	271 (178–443.5)	0.048 ^†^*	−0.122
CD4 count (cell/μL)		18 (8.11–36)	13 (6.74–24)	0.084 ^†^	−0.107
CD8 count (cell/μL)		290 (185.5–494)	228 (145.5–389.75)	0.066 ^†^	−0.113
CD4/CD8		0.06 (0.03–0.11)	0.05 (0.03–0.10)	0.609 ^†^	−0.031
WBC Count (10^9 cell/μL)		4.51 (3.19–6.35)	4.38 (3.61–6.92)	0.567 ^†^	0.034
HIV Viral Load (log10 × copy/mL)		5.02 (4.61–5.52)	5.34 (4.85–5.76)	0.151 ^†^	0.101
PCT (ng/mL)		0.07 (0.04–0.17)	0.15 (0.13–1.24)	<0.001 ^†^***	0.275
ESR (mm/h)		44 (23–68.5)	37 (18–89)	0.631 ^†^	−0.032
EBV	Positive(+)	38 (90.48%)	4 (9.52%)	0.841 ^‡^	−0.012
CMV	Positive(+)	24 (80%)	6 (20%)	0.131 ^‡^	0.109
HBV	Positive(+)	14 (82.35%)	3 (17.65%)	0.548 ^‡^	0.061
ART initiation	Yes	29 (100%)	0 (0%)	0.054 ^§^	−0.116
Days from illness onset to final diagnosis (day)		14 (10–21)	15 (9–28)	0.496 ^†^	0.042

Data are expressed as n (%) or median (interquartile range). Results are based on non-empty rows and columns in each innermost sub-table. ^†^. Significance value of the Mann–Whitney U test. ^‡^. Significance value of the Pearson Chi-square test. ^§^. Significance value of Fisher’s exact test. *. significant at the 0.05 level. ***. significant at the 0.001 level. BDG: β-D-glucan. WBC: white blood cell. CSF: cerebrospinal fluid. HIV: human immunodeficiency virus. PCT: procalcitonin test. ESR: erythrocyte sedimentation rate. EBV: Epstein–Barr virus. CMV: cytomegalovirus.

**Table 3 viruses-16-01333-t003:** In vitro antifungal susceptibility results in HIV-associated cryptococcal meningitis.

Parameters		Outcome		Subtotal	OR	*p* Value
		Survival	Death			
5-FC						
	WT (MIC ≤ 8)	247 (89.82%)	28 (10.18%)	275	2.923	0.357 ^†^
	NWT (MIC > 8)	3 (75%)	1 (25%)	4		
AmB						
	WT (MIC ≤ 0.5)	239 (89.51%)	28 (10.49%)	267	0.777	1.000 ^†^
	NWT (MIC > 0.5)	11 (91.67%)	1 (8.33%)	12		
FCZ						
	WT (MIC ≤ 8)	238 (89.81%)	27 (10.19%)	265	1.467	0.646 ^†^
	NWT (MIC > 8)	12 (85.71%)	2 (14.29%)	14		

Data are expressed as n (%). Results are based on non-empty rows and columns in each innermost sub-table. ^†^. Significance value of Fisher’s exact test. OR: odds ratio. 5-FC: flucytosine. AmB: amphotericin B. FCZ: fluconazole. WT: wild-type definition based on CLSI M57SEd4. NWT: non-wild-type definition based on CLSI M57SEd4.

## Data Availability

The data that support the findings of this study are available on request from the corresponding author. The data are not publicly available due to privacy or ethical restrictions.

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
