# Peer review of "Trends in Clinico-Epidemiological Profile and Outcomes of Patients with HIV-Associated Cryptococcal Meningitis in Shanghai, China, 2013–2023"

_viruses, 2024, doi:10.3390/v16081333_

Round 1

Reviewer 1 Report

Comments and Suggestions for Authors

Zhao and colleagues conducted a retrospective observational study at a single center focusing on patients with AIDS and cryptococcal meningitis (CM), examining trends in incidence, mortality, and clinical characteristics over time. CM remains a significant clinical challenge, and observational studies play a crucial role in updating epidemiological insights into this disease, particularly when dealing with substantial patient numbers, as observed in this study. However, significant revisions are necessary before considering publication.

The abstract should be refined to include specific numerical findings such as incidence rates and mortality data. It is advisable to reconsider describing the study results as "crucial" (line 25), as they largely confirm existing knowledge about CM.

The introduction requires expansion to provide a more comprehensive overview of the study's background.

In section 2.1, laboratory procedures should be detailed separately from the description of study subjects. Methods should also include specifics on HIV testing, AIDS diagnosis criteria, CM diagnostic protocols

Some data are not reported, such as CSF culture, CSF/blood cryptococcal antigen results, India Ink stain findings, and blood culture. Additionally, HIV-RNA levels at diagnosis and treatment details should be included if available (for example, treatment strategies trends over time could be interesting, considering that, at least in Europe, flucytosine is not readily available).

The statistical analysis section needs clarification. Ensure consistency in reporting and citing statistical methods, such as the use of Spearman correlation, which is referenced in Tables 1-2 but not in the Statistical Analysis section. I do not think that the analysis of CSF pressure trends is useful. Finally, I would report p-values rather than statistical test abbreviations (H or chi-square) for making the results section more clear.

Clarify the reporting of identified pathogens, specifying if they refer to those found in CSF. Irrelevant pathogens should be excluded from the analysis (i.e. Acinetobacter or Klebsiella)

Further discussion is warranted regarding the prognostic significance of PCT levels, despite their statistical significance in predicting mortality. Given the median values in both groups (substantially in the reference range), the clinical implications of this finding are questionable and should be further discussed.

Explain why clinical data were omitted from the univariate analysis and subsequent logistic regression model, addressing potential implications for the study's findings.

Italicize "C. neoformans" on line 85.

On line 172, clarify that reduced hospitalization time does not necessarily indicate improved efficacy in CM treatment. 

Comments on the Quality of English Language

I think English style can be improved.

Author Response

We sincerely appreciate your comments. Please see the attached document with our responses below. Thank you for your thoughtful review.

Reviewer 2 Report

Comments and Suggestions for Authors

It is an interesting study because it provides us with information from a series of 279 cases with cryptococcosis and HIV infection. In China, HIV infection has not been as important a risk factor for cryptococcosis as it has been in the rest of the world. 

Despite the decrease in the prevalence of CM in hospitalized patients, the delay in making the diagnosis persists. However, the mortality of 10.39% is low when compared to the rest of the world; in Africa it is approximately 40-60% and in controlled studies 18-28%.

The decrease in symptoms of intracreanal hypertension (headache and vomiting) in the clinical presentation of the disease in recent years is striking. However, these symptoms are still important in the young population. It is a known fact that clinical manifestations in older adults are less specific. The advanced state of immunosuppression (very low CD4+ cells) of the patients over the years is striking.

The decrease in the number of leukocytes in the CSF and the elevation of procalcitonin were the prognostic factors of mortality found. It is interesting that other factors such as altered consciousness, elevated intracranial pressure, and seizures did not reach statistical significance. In this study, sterilization of the CSF culture was not considered among the prognostic factors.

The authors do not clarify which antifungal therapeutic regimen was used in the patients. We do not know if there was a significant difference between the group of deceased and the group of survivors. Although they describe resistance to the 3 antifungals (amphotericin B, fluocytosine and fluconazole), they did not specify which of the different existing regimens they used.The tables and figures are well prepared and useful. I did not have access to the supplementary material. References are appropriate.

In summary, I recommend the publication of this article because it gives us an updated overview of the situation of meningeal cryptococcosis in Chinese patients infected with HIV. It is necessary to specify the type of treatment used.

Author Response

(The authors gave the same response as above.)

Round 2

Reviewer 1 Report

Comments and Suggestions for Authors

I thank the Authors for the revision of their manuscript which made a significant improvement. However, I have some minor comments.

Line 53-54: I think you mean that the diagnosis was confirmed by serology and subsequently by nuclear acid test (i.e. HIV viral load).

I think that the other identified pathogens reported are not relevant. The topic of this paper is CM in AIDS patients, thus we do not really need to know if Acinetobacter or Klebsiella were isolated somewhere. If you want to emphasize the weakness of the immune system in these patients, I would rather report other AIDS-associated conditions if present.

I appreciate your efforts in expanding the discussion about PCT. However, some data reported refers to bacterial infections outside the CNS in non-HIV patients (reference 24 and 27), which are not really useful to interpret the result of this study (consider not to consider these 2 references). Try to rephrase this part of the discussion, focusing on what you report in lines 277-280 which is reasonable.

I can't see Table S4 in the supplementary material.

Line 277: varied.

Author Response

(The authors gave the same response as above.)
